# Novel RL Approach for Efficient Elevator Group Control Systems

## Abstract

The management of elevator traffic in large buildings is crucial for ensuring low passenger travel times and energy consumption. We optimize the Elevator Group Control System (EGCS) using a novel Reinforcement Learning (RL) approach. Existing methods, including heuristic-based and pattern detection algorithms, often fall short in handling the complex and stochastic nature of elevator systems. This research proposes an end-to-end RL-based approach. A custom elevator simulation environment representing the 6-elevator, 15-floor system at Vrije Universiteit Amsterdam (VU) is developed as a Markov Decision Process (MDP). Key innovations include a novel action space encoding to handle the combinatorial complexity of elevator dispatching, the introduction of *infra-steps* to model continuous passenger arrivals, and a tailored reward signal to improve learning efficiency. Additionally, we explore various ways of adapting the discounting factor to the *infra-step* formulation. We investigate RL architectures based on Dueling Double Deep Q-learning, showing that the proposed RL-based EGCS adapts to fluctuating traffic patterns, learns from a highly stochastic environment, and thereby outperforms a traditional rule-based algorithm.

## 1 Introduction

### 1.1 Problem description

In large structures, complex multi-lift systems cater to the high transportation demands. The system responsible for determining which elevators to dispatch in response to passenger requests is known as the Elevator Group Control System (EGCS). The EGCS plays a vital role in ensuring low wait times and energy efficiency (Fernandez & Cortes, 2015). Traditional rule-based control methods can be efficient in simple environments but fail to adapt to the complexity of modern buildings with varying traffic patterns. Reinforcement Learning (RL) offers a promising alternative for optimizing EGCSs by allowing systems to autonomously learn from experience and adapt to complex environments.

Designing an EGCS to efficiently match passengers with available lifts presents several challenges. First, upon registering a floor button press (hall call), the EGCS must choose from multiple carts with varying speeds, positions, destinations, and passenger loads. It may even dispatch more than one cart to serve a new passenger. Second, the system does not know the new passenger's destination until they enter the elevator and register a car button press inside it (car call). Similarly, it only knows the number of passengers associated with a hall call once they board the elevator and register their destinations. The agent must therefore deal with a high level of stochasticity in environment transitions. Third, traffic patterns fluctuate throughout the day, with morning up-peaks and afternoon down-peaks placing the highest demands on elevator capacity. The system can also undergo long-term changes, such as if a building becomes busier over the years. The agent needs to be robust to handle the varying traffic patterns or long term changes. Lastly, solutions must be computed in real-time.

### 1.2 Rule-based approaches

Traditional elevator control systems are often based on rule-based algorithms (Fernandez & Cortes, 2015). An example is the ETD algorithm (Smith, 2002), which minimizes the total trip time for

every new passenger while considering the slowdown incurred to existing passengers. Another popular rule-based approach is sectorization, where the individual elevators are assigned a specific range of floors and respond to hall calls only within these floors (Kameli & Collins, 1996). Idling floors are selected dynamically, for example based on Particle Swarm Optimization (Li et al., 2007) or Dynamic Programming (DP) in combination with traffic pattern detection (Chan & So, 1995). Despite their utility, these algorithms are often too inflexible to deal with varying usage patterns throughout the time of the day, day of the weeks and other key parameters, limiting their optimization potential (Cortés et al., 2012).

## 1.3 RL IN EGCS

RL is highly effective for control-related tasks, enabling an agent to autonomously learn a robust policy through interaction with an environment. When integrated with Deep Learning, RL allows the agent to recognize patterns in the environment data and make fine-grained decisions. This self-learning ability makes RL adaptable to highly specific scenarios, enhancing its suitability for real-world applications. RL is a well-established paradigm, widely applied in domains such as reservoir operation (Castelletti et al., 2010) and complex game-playing (Vinyals et al., 2019).

RL has been applied to EGCSs in limited capacities. Early works such as Crites & Barto (1998) demonstrated the feasibility of RL in optimizing elevator routing. Interestingly, they framed the problem as a Multi-Agent Reinforcement Learning (MARL) problem by having one separate RL controller per elevator. Given the use of a Q-table, the approach faces issues of scalability.

Recent works have integrated deep learning with RL, such as Wei et al. (2020). The authors formalize their solution as a Single-Agent Reinforcement Learning (SARL) problem and implement an Advantage Actor-Critic (A3C) model. They give the model full access to the group sizes and destination floors of future passengers, which are normally unknown to the system. This creates a performance advantage usually unachievable in the real-world.

Another approach developed recently by Wan et al. (2024) improves upon previous limitations by using a Dueling Double Deep Q-learning agent trained in a discrete-event simulation. The agent is only prompted for an action whenever a passenger arrives in the system or an elevator reaches a new floor. They adapt the discounting factor formulation to be able to deal with varying time in-between steps. The authors developed an end-to-end RL controller, capable of fully functioning as an EGCS in a 20-floor, 4-cart environment. One drawback of their implementation is the design of the action space. They require a decision from the RL controller every time any elevator reaches a new floor while moving. The controller should then choose whether to stop at the floor or continue past it. When stopped, the controller should decide at every new event whether to stay idle, go up or go down. This induces a large and complex action space and a large action density in a learning episode, making the problem difficult.

## 1.4 AIMS AND CONTRIBUTIONS

This paper presents an RL-based EGCS to address the shortcomings of the current approaches, with both methodological and practical contributions.

Our contributions can be summarized as follows:

### 1.4.1 METHODOLOGICAL CONTRIBUTIONS

- We formulate the elevator routing problem as an MDP to allow for RL interaction,
- We propose the use of *infra-steps* to adapt to the continuous nature of passenger arrivals, while allowing the RL agent to model and learn from the system dynamics occurring between discrete decision points,
- We design and test a specialized discounting strategy tailored to the *infra-step* formulation, enabling the RL agent to account for variable time intervals between actions, thus improving the overall learning and decision-making process,
- We design two RL architectures based on Dueling Double Deep Q-learning and compare them.

### 1.4.2 PRACTICAL CONTRIBUTIONS

- We develop a realistic elevator simulation environment that closely mirrors the 6-elevator, 15-floor system at Vrije Universiteit (VU) Amsterdam, using data collected in June 2023. This simulator allows us to validate and test the performance of the proposed RL-based EGCS reliably,

- We introduce a novel action space encoding, specifically tailored to the elevator dispatching problem, which is designed to side-step the combinatorial complexity typically associated with this problem domain,

- We design the reward signal to provide more frequent feedback to the RL agent, with the goal of improving learning efficiency and accelerating convergence by addressing the issue of sparse rewards,

- The RL-based solution we propose outperforms a modern rule-based system in terms of passenger travel time, offering a promising alternative for real-world EGCS implementation.

Through these contributions, our work advances the methodological understanding of how RL can be adapted to the continuous and complex nature of elevator systems. It also demonstrates the feasibility of an RL-based EGCS in real-world applications.

## 2 METHODOLOGY

### 2.1 ELEVATOR SIMULATION

Floor and elevator button presses were recorded at the VU Amsterdam in June 2023 and passenger arrival times and destinations were reconstructed based on that data. Hall calls were matched to the relevant car calls in order to deduce the destination of a passenger group. As there was no way of knowing how many people were behind one car call, we estimated the group size per car call by sampling from a geometric distribution as had been done in previous research (Sorsa et al., 2021). We define the distribution parameters per hour based on Sorsa et al. (2021), as they provide distribution parameters that were stable across a wide range of building types and locations. A summary of traffic patterns that were recorded can be seen in Figure 1.

### 2.2 DISCRETE-EVENT ENVIRONMENT

### 2.3 MARKOV DECISION PROCESS

An RL agent learns a policy that maximizes cumulative rewards by interacting with its environment. The environment is formulated as a Markov Decision Process (MDP). An MDP is formalized as a tuple $(S, A, P, R, \gamma)$, where $S$ is the set of states produced by the environment and observable by the agent, $A$ is the set of actions available to the agent, $P(s'|s, a)$ is the probability of arriving in state $s'$ after taking action $a$ in state $s$. $R(s, a)$ is the reward of taking action $a$ in state $s$, and $\gamma$ is the discount factor.

An agent interacting with an MDP aims to find a policy $\pi$ that maximizes the expected discounted reward $G^\pi$.

$$G^\pi = \mathbb{E}\left[\sum_{t=0}^{\infty} \gamma^t R(s_t, a_t)\middle| \pi\right] \tag{1}$$

where $t$ represents all the timesteps-to-be until the terminal state, $R(s_t, a_t)$ is the reward obtained for taking action $a_t$ in state $s_t$, and the policy $\pi$ determines the actions.

### 2.3.1 ACTION SPACE

A key challenge in optimizing EGCS using RL is designing a suitable action space to be used in the MDP.

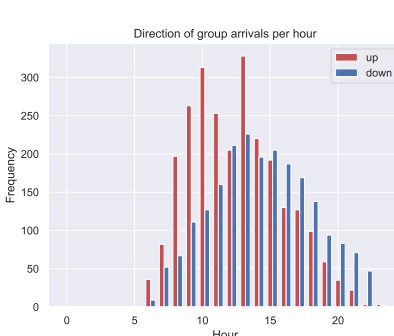

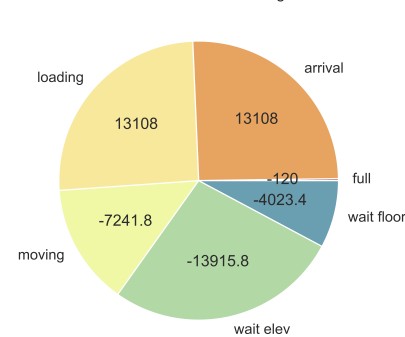

Figure 1: Distribution of travel directions. The morning shows a majority of upward travel, lunchtime is both directions, and the evening has more downward travel, although the peak is not as pronounced as the morning up-peak.

Figure 2: Construction of the reward in a total environment run by a trained RL agent.

Wei et al. (2020) and Wan et al. (2024) define the action space for every elevator to either continue or stop at the next floor, leading to an action space of $2n$, where $n$ is the amount of elevators. This reduces the action space, but the action density is high as the system is prompted for an action every time any elevator reaches a new floor.

Alternatively, the EGCS can act on a higher level by matching each active hall call to an elevator, and the lower-level movement decisions are then delegated to a module that makes the carts stop at every floor for which the cart has a hall call assigned. However, in a multi-elevator system, each elevator can respond to each hall call, leading to significant combinatorial complexity in the number of possible actions. In a system with $n$ elevators and $m$ active hall calls, there are $n^m$ possible assignments (Hamdi & Mulvaney, 2007).

In order to avoid both problems, we formalize the action space differently. We only prompt the EGCS for an action when a new passenger or passenger group enters the system and registers a hall call. The EGCS then decides which elevator(s) to send to the new hall call. The hall call floor is then added to the selected elevator(s)'s destination queue, and each elevator then independently works on emptying their destination queue automatically, without requiring further input from the RL agent. Passenger expectations dictate that the elevator should always stop at a floor for which a car call is registered and it should empty its destination queue in the direction of travel before reversing. These constraints imply that there is only one course of action possible, and there is no need to make further action decisions.

### 2.3.2 INFRA-STEPS

Traditional RL frameworks typically operate in time-discrete MDPs. Because of the asynchronous nature of elevator arrivals in our environment, there is a variable time in-between decision moments where the agent is prompted for an action. We implement a discrete-event environment and introduce the concept of *infra-steps*, defined as intervals of 0.1 seconds during which the system refreshes elevator positions and processes events. When an event appears, the environment fetches the present state and prompts the agent for an action. There can therefore be a variable number of *infra-steps* between steps. However, the agent still only interacts with the environment at every step, as in the basic formulation of discrete-time RL. This mechanism is illustrated in Figure 3.

### 2.3.3 DISCOUNTING FACTOR

An important implication of the *infra-step* formulation is determining how to address discounting regarding the variable time between the steps. One option is to apply discounting on the *infra-step* level, which means the reward is summed up at every *infra-step* and discounted appropriately. The resulting modified Bellman equation is the following:

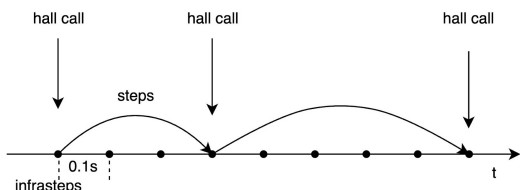

Figure 3: Discrete-event formulation of the environment. The simulation updates every 0.1 seconds and verifies if a hall call has been registered. The environment processes automatic behaviour. Only when a hall call is registered does it return the state to the agent and prompt it for an action.

$$V(s_t) = \sum_{t=0}^{N-1} \gamma^t * R(s_t, a_t) + \gamma^N * V(s_{t+N}) \tag{2}$$

where $R(s_t, a_t)$ is the reward obtained at *infra-step* $t$ after taking action $a_t$ in state $s_t$, and $t$ is the *infra-step* indicator. The agent cannot change its action in-between two states. $\gamma$ is the discount factor and $V(s_t)$ and $V(s_{t+N})$ are the values of state $s_t$ and next state $s_{t+N}$, respectively.

Another option is to entirely abstract away from the variable timing between steps and discount every next step by the same discount factor, regardless of the *infra-step* amount. This means that even though the environment is a discrete-event scenario, the agent sees it as a discrete-time environment. The reward then simply becomes the sum of rewards acquired between $s_t$ and $s_{t+N}$. The Value function is then the traditional Bellman equation:

$$V(s_t) = \sum_{t=0}^{N-1} R(s_t, a_t) + \gamma * V(s_{t+N}) \tag{3}$$

There are limited theoretical foundations for such a problem, and one notable article that implements a discrete-event formulation (Wan et al., 2024) employs the variable discounting approach without further theoretical justifications. We tested both approaches in practice and compared them experimentally.

### 2.3.4 ENVIRONMENT STATE AND REWARDS

To build a state $s$, we selected features from the elevator environment that were most relevant to the RL agent. The state $s$ was composed of the following elements:

1. The floor the current hall call has been registered at (0 - 15)

2. The direction of the current hall call, -1 when going down, 1 when going up

3. Time, as a fraction of the day (0 - 1)

4. Day of the week, as an integer (1 - 7)

5. For every elevator, an encoding of its position in meters, divided by the total building height (0 - 1)

6. For every elevator, its ETD score

7. Speed for every elevator in meters per second, negative when going down, positive when going up (-2.5 - 2.5)

8. Current weight in every elevator, in kilograms.

The ETD score was a measure of the general busyness of every elevator as designed by the ETD algorithm (Smith, 2002) (see section 2.4.1)

The primary reward signal the agent receives from the environment is a negative reward for each passenger either waiting on a floor or traveling in an elevator. The system therefore aims to clear

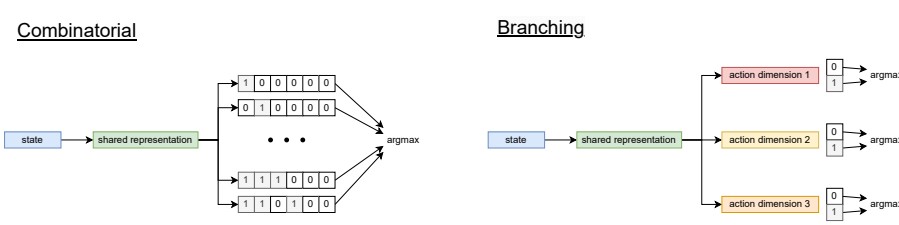

Figure 4: Comparison of both RL agent architectures.

all passengers to minimize negative rewards. However, there can be a large amount of steps in busy times in between an action taken by the agent and the moment the passenger exits the system. To make a more direct connection between actions and rewards, we introduced extra reward signals that the agent got earlier:

1. *Wait floor* and *Wait elev* are the penalties incurred at every *infra-step* for every passenger currently waiting at a floor or travelling in an elevator, respectively.

2. *Loading* is the reward obtained when a passenger boards an elevator.

3. *Arrival* is the reward when a passenger is dropped off at their destination.

4. *Moving* is the penalty incurred at every *infra-step* for every moving elevator. This is included to give the EGCS a sense of energy preservation.

5. *Elevator full* is the penalty incurred when an elevator is full and cannot accept a passenger while serving a hall call.

The rewards were balanced manually against each other. The sum and relative importance of rewards of one environment run can be seen in Figure 2. The state was normalized per element before being passed to the neural network. Mean and standard deviation of the training data were used for normalization. Further details on the state elements and rewards can be consulted in the Appendix A.1.

### 2.4 AGENTS

#### 2.4.1 BASELINE AGENT

To assess the performance of our RL algorithm, we implemented the rule-based ETD algorithm developed by Smith (2002). The ETD algorithm aims to minimize the total time impact incurred by serving a new passenger in the system. It minimizes the time to serve the passenger itself while also considering the impact on other passengers' journey time if a certain elevator were to pick up the passenger. This is formalized with the following equation:

$$Cost_e = ETD_e + \sum_p Delay_{e,p}$$

where $ETD_e$ is the Estimated Time to Destination in seconds for elevator $e$ and $Delay_{e,p}$ is the delay in seconds incurred to passenger $p$ currently being served by elevator $e$. The cost for elevator $e$ is the sum of the time it will take to serve the passenger plus the delay incurred to all other passengers $p$ currently served by elevator $e$. The elevator $e$ with the lowest total cost is chosen.

This algorithm is effective because it directly minimizes the system's target metric: the total travel time per passenger. The algorithm is still in use nowadays in many ThyssenKrupp elevator systems (Latif et al., 2016).

We further validated our baseline agent by comparing it to other types of baseline algorithms. See Appendix A.2 for comparisons.

### 2.4.2 RL AGENT

At every step where a hall call is registered, the environment provides the current state and the agent is then responsible for deciding which elevator(s) it will send to the hall call.

We created several variations of an RL agent based on a Dueling Double Deep Q-learning architecture.

1. Combinatorial Action Space: The RL agent computes the Q-values for all combinations of elevators responding to the new hall call, from one to three elevators. This results in an action space of $C(6, 3) + C(6, 2) + C(6, 1) = 41$ outputs. The chosen action is the argmax over all outputs.

2. Action Branching: Inspired by the work of Tavakoli et al. (2018), we employ an action branching strategy. The state is processed commonly by a shared NN, and branches off to six individual NN's, where each elevator makes an independent binary decision (respond or not to a call). This decomposes the high-dimensional action space into multiple lower-dimensional sub-spaces, and results in an action space of $6 * 2 = 12$ outputs. The chosen action is the aggregate of the argmax of every branch.

Both architectures are shown in Figure 4.

## 3 RESULTS AND DISCUSSION

RL agents were trained on 10M steps and evaluated periodically on the validation environment. The best performing agent was selected and tested 20 times in the test environment to estimate final performance. Training details are provided in the Appendix A.3.

### 3.1 AGENT DESIGN AND COMPARISON TO BASELINE

We tested the difference in the NN architectures employed. We compared the Combinatorial architecture, where every action combination is a single output, to the Branching architecture, where separate outputs control one elevator each. We also compared both algorithms to the baseline ETD algorithm. Figure 5 shows experimental results.

Both versions of the RL agent beat the baseline by about 2 seconds per passenger, or approximately 10%.

The Combinatorial agent is largely superior to the Branching agent. Interestingly, the average wait time per passenger is similar, although the Branching agent spends more energy. Looking at the number of elevators sent per call, we see that the Branching agent often sends fewer than one elevator to a call on average, meaning it sometimes sends zero elevators to a hall call. It thereby incurs a large negative reward every time. It is unable to coordinate its branches well enough to avoid this situation. As coordination between multiple sub-actions is crucial in this problem, the agent benefits from having one central decision-making module, outweighing the disadvantage of more complex input and output spaces.

Further experiments on the agent action space design were conducted, as well as additional tests to assess the robustness and adaptability of the agent by running it on a busier environment than it was initially trained on. The results of these experiments can be seen in Appendix A.4 and A.5, respectively.

### 3.2 DISCOUNTING FACTOR

We compared the fixed discounting scheme to the variable discounting scheme as explained in Equations 2 and 3. Figure 6 shows experimental results.

As is apparent in the average rewards obtained and the average passenger waiting times, the fixed discounting approach works best for two main reasons.

First, the number of *infra-steps* contained in a step is highly variable. During the daytime, arrivals can happen within one *infra-step* of each other or even on the same *infra-step*, whereas during the

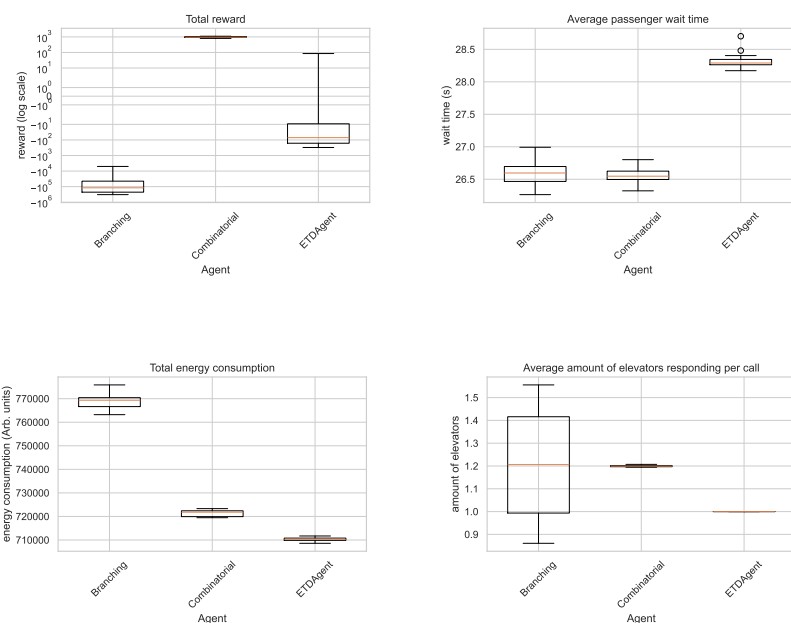

Figure 5: Comparison of the Branching and the Combinatorial architecture to baseline. The reward is in log scale. Error bars represent SD for 20 runs on the test environment.

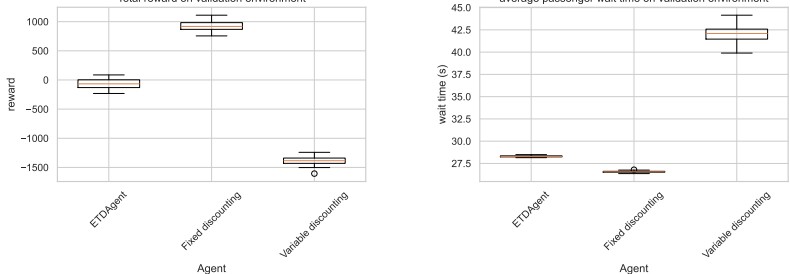

Figure 6: Comparing fixed to variable discounting schemes. Bars represent SD on 20 test environment runs.

night, there might be hours between two arrivals, which results in tens of thousands of *infra-steps*. On long inter-arrival times, the value of the next state quickly drops to a small value in Equation 2, making for a very unstable learning problem. Further details on this problem can be consulted in the Appendix A.6. Second, the problem balances itself out quite well: at low-peak hours, where the inter-step time is long, nothing happens, and the system receives no reward, whereas in peak times, the system receives very frequent rewards. By disregarding the inter-step length, the average rewards remain similar in transitions, regardless of the time elapsed between the two decision points. This is illustrated in Figure 7, where the average reward remains relatively uniform through step lengths. The reward obtained in a transition is largely uncorrelated to the step length, making the fixed discounting scheme more effective in practice.

## 4    LIMITATIONS AND FUTURE DIRECTIONS

To actually implement the trained EGCS in the VU building, some limitations must be addressed.

Specific strategies are required for handling unique situations, such as when an elevator is out of service due to maintenance or emergency mode such as fire. The RL EGCS could be integrated with a rule-based complement to deal with these specific scenarios.

An important area for performance improvement is incorporating the ability to revise decisions when they become sub-optimal. Currently, once an elevator is dispatched to a hall call, it remains committed to that choice, even if later hall calls could make the initial decision less efficient. Utgoff & Connell (2011) propose a method where decisions are continuously recalculated using local search, allowing for substitution if a better solution is found. However, due to the computational complexity of exploring the local solution search space, their solution has limited success.

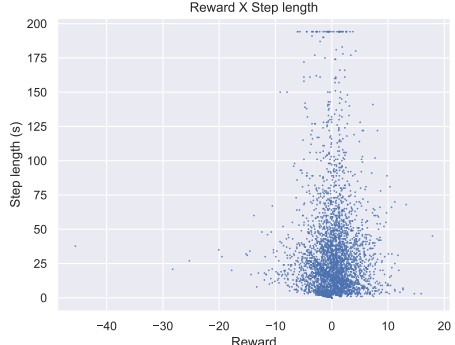

Figure 7: Distribution of step length and reward per step. Data was cut off at 190 for better visualization, hence the ceiling data points (cutoff was only for plotting).

Even though the RL agent performs well as-is, it would be a notable improvement to provide it with a framework where it can revisit certain decisions in situations where it would be beneficial.

A further area of interest is looking further into the trade-off of passenger travel time and energy consumption. In our case, because of the balance of the sub-rewards, the agent implicitly learns to prioritize passenger travel time over energy consumption. As a result, the EGCS achieves lower passenger travel times than the ETD baseline but at a higher overall energy consumption cost. One advantage of our approach is that it is trivial to shift the balance toward energy saving or passenger travel time by changing the weights of the rewards in the environment. However, the agent would need to be re-trained to learn a new reward balance.

Once these limitations are overcome, one would be able to assess the performance of the approach in a real-life setting and to verify its ability to bridge the reality gap.

## 5    CONCLUSION

In this paper, we addressed the problem of optimizing complex elevator dispatching using a novel RL approach. By formulating the problem as an MDP, we captured the inherent uncertainties and complex nature of elevator systems, particularly with the introduction of *infra-steps* to simulate continuous passenger arrivals.

Methodologically, we gained valuable insights by comparing two discounting schemes: fixed and variable. The fixed discounting strategy proved more stable and effective in managing varying time intervals between actions. Additionally, we compared two RL agent variants: the branching and combinatorial agents. Our results show that the combinatorial architecture outperformed the branch-

ing strategy, leading to more efficient decision-making. The innovative encoding of the action space allowed us to overcome the typical combinatorial complexity in this domain.

On the practical side, using a custom-built simulation environment of a 6-elevator, 15-floor system at VU Amsterdam, the proposed RL-based solution demonstrated superiority over a modern rule-based system. The RL agent, utilizing the Dueling Double Deep Q-Learning algorithm, was able to significantly reduce passenger travel times by efficiently adapting to complex traffic patterns. These promising results underline the potential for practical implementation of RL-based control for real-world elevator systems, particularly in environments with complex, stochastic traffic patterns that can be optimized from data.

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

# A   APPENDIX

## A.1   DETAILS ON ENVIRONMENT DESIGN

The environment and agents were implemented in Python 3.11.

The weight of every new passenger that was included in the state, was sampled from a normal distribution with mean 75 and standard deviation 10.

The weights of the rewards that the agent received are summarized in Table 1.

| Reward | Value | When |
|---|---|---|
| Movement penalty | -0.01 | At every *infra-step*, for every elevator currently moving |
| Waiting penalty | -0.01 | At every *infra-step*, for every passenger currently waiting on a floor or in an elevator |
| Elevator full penalty | -10 | Once, when an elevator stops at a floor to service a call but it is full |
| Arrival reward | 2 | Once, when a passenger disboards an elevator |
| Loading reward | 2 | Once, when a passenger boards an elevator |
| Zero elevators responding penalty | -100 | Once, when the agent sends 0 elevators to a hall call. Only relevant for the Branching agent |

Table 1: Rewards used to train the agent, their values, and when they are received by the environment.

## A.2   ADDITIONAL BASELINES

To verify the performance of our Baseline agent, we compared it to additional, simpler baselines.

Hereunder are the simple baselines used:

1. **Random Agent** This agent assigns a random elevator to every hall call
2. **First Agent** This agent assigns elevator 1 (out of 6) to every hall call.
3. **Closest Agent** This agent assigns the elevator that is physically closest to the hall call location, regardless of whether it is moving or how many passengers it is currently serving.
4. **Sector Agent** This agent splits the building into six equal zones, and every elevator responds to hall calls within its zone. This idea is still used in many elevator systems, although in a more sophisticated configuration (Crites & Barto, 1998).
5. **Least Busy Agent** This agent assigns the hall call to the agent with the smallest number of destinations in its current queue (hall calls + car calls).

We ran all baseline algorithms for 20 iterations on the test environment to get a reliable estimate of average performance. Figure 8 shows that most agents perform better than the absolute baseline of sending a random elevator to the hall call. The ETD baseline performs best overall. The LeastBusy agent performs comparably to the ETD agents, although it is slightly less efficient. This is likely because the length of the destination queue of an elevator is strongly related to its ETD score. Surprisingly, the Sector agent performed worse than the Random agent. This is likely because some floors see more traffic than others, and assigning only one elevator to those busy floors would be comparable to always choosing the busiest elevator to respond to those calls. Modern Sector algorithms adjust the zones dynamically based on observed traffic patterns and can have multiple elevators per zone (Al-Sharif et al., 2019), avoiding this problem. Our implementation of the agent is therefore too simplistic to reflect the performance of a modern Sector agent.

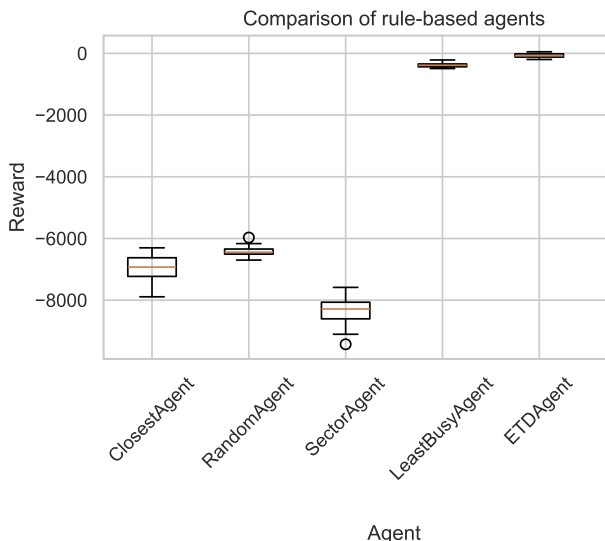

Figure 8: Total rewards of baseline agents on the test environment. Error bars represent the standard deviation (SD) on 20 iterations of the test environment.

## A.3 DETAILS ON AGENT TRAINING

The Combinatorial agent has three hidden fully-connected layers of size 128, 512, and 256 neurons, respectively, with ReLU non-linearities. The total input size is 28, while the maximum output size is 41 in the case of up to three elevators being allowed to respond to a hall call. The Branching agent has the same main body, but the last layer before the output splits into six distinct branches of 128 neurons each before proceeding to the output layer. This NN has 28 input neurons and 12 output neurons.

The optimizer used for these NNs is AdamW, with a learning rate of 5e-4. We also apply a Cosine Annealing scheduler to the learning rate, which reduces the learning rate throughout training from 5e-4 to 0. This is done to stimulate exploration at first and exploitation later in the training. We use the Huber loss function.

All further run parameters are set as described in Table 2.

For experiments, RL agents were trained on 10M steps of the train environment. The environment was composed of 27684 steps, so the agent saw every transition $\approx 361$ times. The mean and standard deviation of state features of the training environment were used to normalize the state features of the training, validation and test environment. The agent was evaluated 30 times at regular intervals on the validation environment. The best agent on validation was retained as final agent. The selected agents were then ran 20 times on a full run of the test environment to obtain an unbiased estimate of the actual performance.

An example of learning curves of a full training run can be seen in Figure 9.

## A.4 ACTION SPACE SIZE

We compared versions of the agent that could send up to one, two, or three elevators to a hall call. The advantage of being able to send less elevators to a hall call is a reduced action space, at the cost of a less expressive action space. The action space in case of being able to send one elevator max is $C(6,1) = 6$. In case of two elevators max, the action space is $C(6,2) + C(6,1) = 21$. In case of three elevators max, the action space is $C(6,3) + C(6,2) + C(6,1) = 41$ outputs. The agent is never allowed to send zero elevators so that action was not included. We did not allow more than three elevators to respond to a hall call as this would make the output size too large and would also not be realistic in practice.

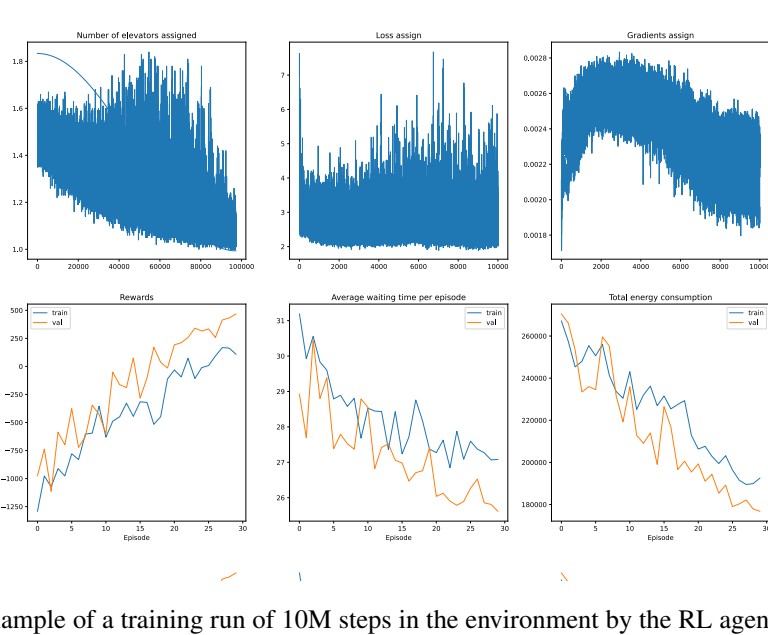

Figure 9: Example of a training run of 10M steps in the environment by the RL agent. The number of elevators sent to a call is noisy but decreases, while the loss does not obviously decrease through training. The reward obtained on the training environment increases through training, as well as the reward obtained of the validation environment. The resulting average passenger waiting times and energy consumption decrease, both on the training and on the validation environment.

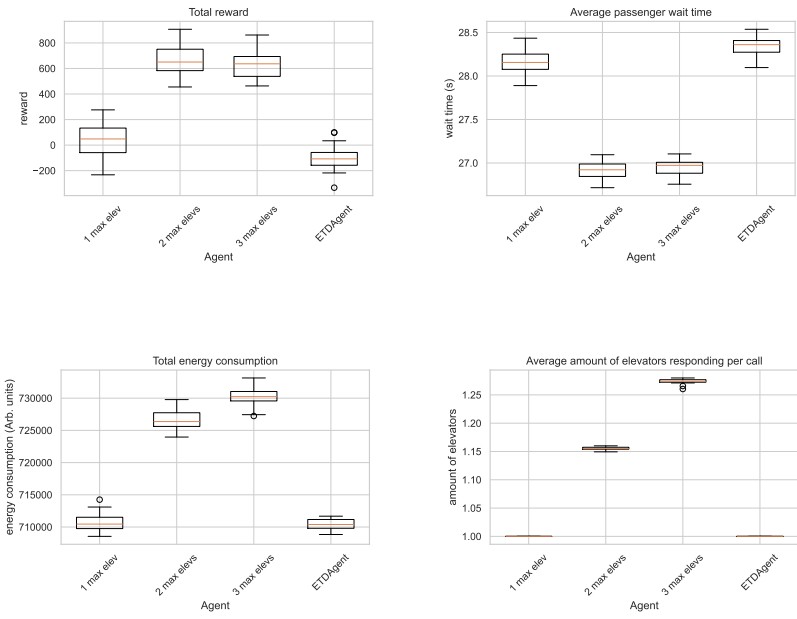

Figure 10: Effect of quantity of elevators sent for one hall call. Error bars represent SD on 20 test environment iterations.

| Parameter | Value |
|---|---|
| Batch Size | 32 |
| Learning Rate | 5e-4 |
| Learn interval | 10 |
| Target network update interval | 300 |
| Discount factor | 0.95 |
| Training steps | 10.000.000 |
| Replay buffer size | 10.000 |
| Replay buffer initial size | 10.000 |
| Epsilon start | 1 |
| Epsilon end | 0.1 |
| Epsilon decay | Exponential |

Table 2: Parameters used for training. Learn interval is the number of steps between each forward-backwards pass on the NN on a batch. The target network update interval is the interval at which the target network updates itself with the online network values.

Figure 10 shows that the RL agent with one elevator performs slightly better than the baseline already, although the difference is slight. It can obtain similar passenger wait times for a similar total energy consumption. This confirms that the agent can extract useful patterns from the environment, learn from it, and match the baseline's performance.

Figure 10 also shows that the agent benefits from being able to send more than one elevator to a hall call, as hypothesized. It achieves a higher reward than the one-elevator variant and the baseline due to a reduced passenger wait time. However, it uses more energy to attain that goal. This finding is central to our research, as it validates that the RL agent effectively and autonomously learns to extract relevant information from the environment. It can then make efficient decisions to solve the complex EGCS routing problem with its various dynamic sub-parts.

Surprisingly, allowing the system to send up to three elevators to a hall call slightly negatively impacts the performance. It matches the average wait time of the two-elevator system but requires more energy to do so, resulting in a lower reward. Figure 10 shows that the three-elevator system sends more elevators per call than the two-elevator variant, increasing the energy spent. The lower performance of the three-elevator system is notable, as it technically can execute all the actions of the two-elevator system. Therefore, it should be able to attain at least the same performance. However, the extra complexity in the action space (21 vs 41) likely makes it more challenging to train in the same allotted training time, hence the inferior performance.

## A.5 ADAPTABILITY TO BUSIER SCENARIOS

To test if the trained RL EGCS would be able to adapt to a building that grows busier over time, we implemented an adapted busier version of the environment. The mean hall call group size was multiplied by 1.5 and 2.0 to create busier scenarios. We compared the performance of the baseline agent to our RL agent trained on the original environment to assess whether the agent could adapt to a scenario it had not seen during training.

Figure 11 shows experimental results. The baseline agent and the RL agent collect more rewards in one iteration of the 1.5x test environment, as there are more opportunities to pick up and drop passengers, thereby earning more rewards. In the 1.5x and 2x cases, the RL agent trained on the original 1x environment surpasses the baseline agent's performance, even though it was not trained on these specific scenarios. The baseline agent collects more rewards on the 1.5x scenario, but its performance drops on the 2x scenario. The RL agent's performance and energy efficiency drop in busier scenarios but less so than the ETD agent's performance. The average passenger wait time grows with busyness but is always inferior to the ETD agent. This shows the flexibility of the policy that the agent has learned in training and is a good indication of the robustness of the resulting EGCS.

Interestingly, the RL EGCS sends fewer elevators per call in the 1.5x setting and even less in the 2x setting. This suggests that the system learnt that sending more than one elevator per call is only

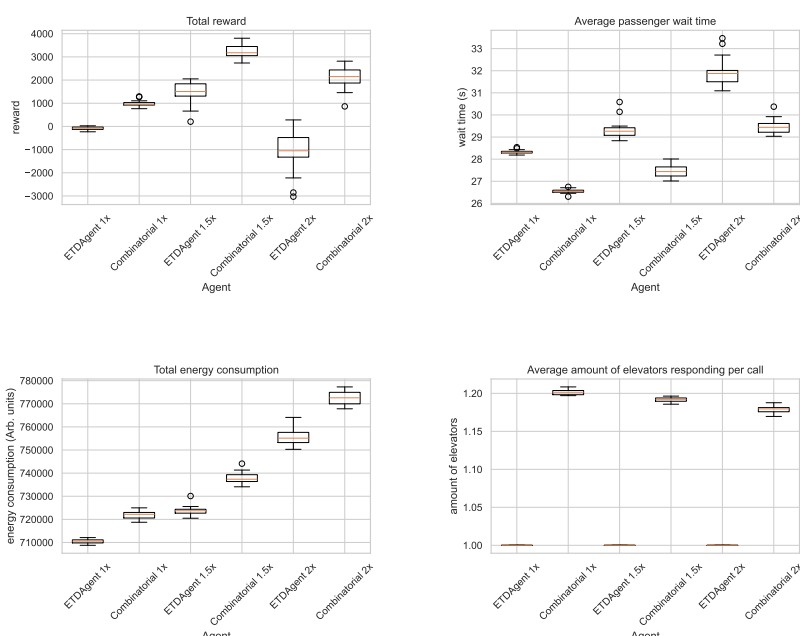

Figure 11: Performance of the RL agents in a 1.5x and 2x busier scenario. The RL agent is trained on the original environment. Error bars represent SD on 20 iterations on the test environment.

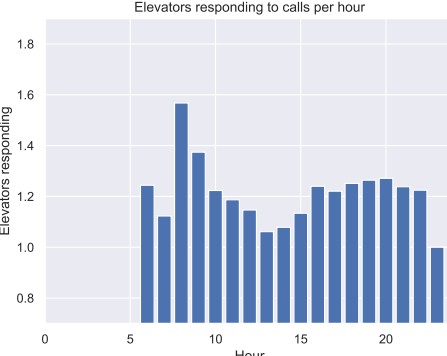

Figure 12: Average number of elevators responding to a hall call per hour. Hours 0-5 are empty as there are no arrivals at these times in the test data.

beneficial in less busy times, as this ensures as many elevators as possible remain available for calls in the immediate future during busy periods. Figure 12 corroborates this idea, as it shows that the RL agent sends fewer elevators per call during lunchtime, which is the busiest period, as seen in Figure 1. The late-night peak is also surprising, as we expect no need for more than one elevator per call during quiet hours. As for the Zoning agent, the agent probably does not have enough data points on these hours to make a reliable policy.

## A.6 FIXED VS. VARIABLE DISCOUNTING PROBLEM

To illustrate the issue with the variable discounting approach, we calculated that the average number of *infra-steps* per decision step is 328 in the training environment. For example, if we want to set the average discount factor between steps as 0.95, the *infra*-step level discount factor must be $x^{328} = 0.95$, or $x \approx 0.99999$. However, during peak hours, where the inter-step time is often one

*infra-step*, the discount factor of the next state is close to 1. The large range in step sizes induces instability when using the variable discounting approach.

