# OpenReview forum: "Novel RL Approach for Efficient Elevator Group Control Systems"
_ICLR.cc/2025/Conference — ICLR 2025 Conference Withdrawn Submission_

### Official Review · Reviewer_iYfR · 2024-11-01

**Soundness:** 2
**Presentation:** 2
**Contribution:** 2
**Rating:** 3
**Confidence:** 4

**Summary:**

The paper studies the group elevator control problem by introducing RL.
The paper is clearly written and easy to follow,
however contribution is minor.

**Strengths:**

The topic of the paper is interesting. The paper is clearly written and easy to follow.

**Weaknesses:**

The contribution of the paper is minor in the sense that the details of the key elements proposed method are missing. For example the deep neural networks are not given. The other limitation is that the quality of the simulation model used for training the elevator group control algorithms is not clear.

**Questions:**

What are the deep network models used in the proposed RL?
How does the proposed RL different from other possible solutions such as rule based algorithms?

---

### Official Review · Reviewer_fNZ9 · 2024-11-04

**Soundness:** 2
**Presentation:** 3
**Contribution:** 2
**Rating:** 5
**Confidence:** 5

**Summary:**

The paper proposed an RL algorithm for elevator group control systems. The authors proposed a new action space to handle the combinatorial complexity of elevator dispatching. The infra-steps are proposed to handle continuous passenger arrivals. Overall it is a good application paper for RL, the writing is clear and the modification is reasonable in practice.

**Strengths:**

1.	The authors focus on a very practical and meaningful real-world problem, which should be encouraged in the RL community.
2.	The writing is very clear. Especially, the authors explained many definitions the elevator control very well.
3. The proposed new action space and infra-steps look simple but effective, which might benefit the empirical RL research very much. The significance is beyond the elevator group control.

**Weaknesses:**

1. The notations are sometimes confusing. For example, in equation (1) $G^\pi$ is a conditional expectation, which is not correct. $\pi$ is not a random variable. $\pi$ is a function and will change the state-action distribution. A common practice is to write $G$ as a function of $\pi$.

2. The contribution of infra-step is not very clear. The empirical results have shown that the fixed discounting works better than the variable discounting.

**Questions:**

The questions are mainly about the infra-steps proposed.

1. What are the contributions of the proposed infra-steps (see weakness 3)?

2. What are the motivations and theoretical insights for the variable discounting factor/infra steps? As the discounting factor is just for contraction mapping for the convergence, then it is okay to be either variable or constant as long as it is smaller than 1.

I will consider increasing my score if the questions are properly answered.

---

### Official Review · Reviewer_PQVi · 2024-11-07

**Soundness:** 2
**Presentation:** 2
**Contribution:** 2
**Rating:** 3
**Confidence:** 4

**Summary:**

This paper addresses the optimization of complex elevator dispatching using a novel reinforcement learning (RL) approach. By modeling the problem as a Markov Decision Process (MDP) and introducing infra-steps to simulate continuous passenger arrivals, the authors capture the inherent uncertainties and complexities of elevator systems.
The paper compares fixed and variable discounting strategies, finding that the fixed approach provides greater stability and effectiveness in managing varying time intervals between actions. Additionally, the research evaluates branching and combinatorial RL agent architectures, demonstrating that the combinatorial architecture leads to more efficient decision-making.
Empirical results show that the proposed RL-based solution outperforms modern rule-based systems in a simulated environment with six elevators and fifteen floors. The RL agent utilizes a Dueling Double Deep Q-Learning algorithm to efficiently adapt to complex traffic patterns, significantly reducing passenger travel times. These promising findings underscore the potential for practical implementation of RL-based control in real-world elevator systems.

**Strengths:**

The paper demonstrates significant strengths through its innovative approach to elevator dispatching using a novel reinforcement learning (RL) framework.
By introducing infra-steps to simulate continuous passenger arrivals and formulating the problem as a Markov Decision Process (MDP), it effectively captures the complexities of elevator systems.
The comprehensive comparison of fixed and variable discounting strategies, along with the exploration of branching and combinatorial RL architectures, reflects methodological rigor and originality.
The research is presented with clarity, supported by detailed diagrams and equations, which enhance understanding. Furthermore, the study has considerable significance,
offering a practical reduction in passenger travel times and bridging theoretical and practical applications in real-world elevator management systems.

**Weaknesses:**

1. Experimental Comparison : The paper only compares the proposed method against the classical ETD algorithm. It does not include comparisons with recent RL-based approaches, making it difficult to evaluate the method's novelty and effectiveness in the broader RL research context.
2. Experimental Setup : The experiments are conducted using a single dataset, which limits the capacity to demonstrate the method's adaptability to diverse scenarios or environments. Testing across various conditions would better demonstrate robustness and versatility.
3. Results Clarity : The results do not clearly show how the proposed algorithm outperforms previous methods. Adding more detailed analysis and comparison metrics would help elucidate the specific advantages.
4. Action space : The paper mentions a significant reduction in action space design but does not offer direct comparisons with previous algorithms. Including these comparisons would strengthen the explanation and highlight improvements.

**Questions:**

1. How does the reduction in action space compare specifically to other methods in terms of computational efficiency and decision-making effectiveness?
2. What specific metrics or analyses detail the advantages of your algorithm over existing methods?
3. Could you provide a comparison of your method with recent RL-based approaches for elevator dispatching or similar dynamic scheduling problems?
4. Has the method described in the paper been tested in real-world scenarios, and does it encounter any latency issues? How does it handle unexpected situations such as elevator malfunctions or occupancy?

---

### Official Review · Reviewer_PFQs · 2024-11-09

**Soundness:** 1
**Presentation:** 1
**Contribution:** 1
**Rating:** 1
**Confidence:** 3

**Summary:**

This paper introduces a reinforcement learning (RL) approach to optimize elevator group control systems (EGCS).
By incorporating infra-steps to model continuous passenger arrivals,
the RL-based method outperforms traditional rule-based systems in minimizing passenger wait times.
The study demonstrates significant potential for real-world applications in dynamic, high-traffic environments.

**Strengths:**

- This feature, which models continuous passenger arrivals, creates a learning environment for the RL agent that mirrors real-life complexities.
- The paper's approach is designed to avoid combinatorial complexity, ensuring efficient decision-making through a well-structured action space. This design choice provides a sense of relief about the model's efficiency.
- The simulation design is based on the actual data set.

**Weaknesses:**

- This paper is still in the stage of considering the use of reinforcement learning, and the comparison with existing methods is insufficient.

**Questions:**

Is this technology intended for elevator control?
Please clearly state the differences compared to what has already been achieved with existing control technology.

---

### Note · Authors · 2024-11-20

I have read and agree with the venue's withdrawal policy on behalf of myself and my co-authors.